# Obstructive sleep apnea and outcomes in acute pulmonary embolism: A large-scale database study

Saud Alawad[1], Nawaf Al-Saeed[1], Ahmad Jarrar[2], Sijin Wen[3], Sunil Sharma [2]*

**1** Department of Medicine, West Virginia University School of Medicine, Morgantown, West Virginia, United States of America, **2** Division of Pulmonary, Critical Care and Sleep Medicine, West Virginia University School of Medicine, Morgantown, West Virginia, United States of America, **3** Department of Epidemiology and Biostatistics, West Virginia School of Public Health, Morgantown, West Virginia, United States of America

* sunil.sharma@hsc.wvu.edu

## Abstract

### Study objectives

To evaluate the impact of obstructive sleep apnea (OSA) on outcomes of acute pulmonary embolism (PE). The primary objective was to compare 30-day mortality and incidence of cardiac arrest in patients with known OSA who developed acute PE versus those with acute PE without OSA.

### Methods

A retrospective cohort study was conducted using the TriNetX global health research network. Two cohorts were defined: adult patients with OSA and acute PE, and adult patients with PE without OSA. Propensity score matching was used to address demographic and comorbidity differences. Data from January 1, 2013, to December 1, 2024, were analyzed, with a one-month follow-up for secondary outcomes.

### Results

The OSA group (n = 3,547,220) had higher PE incidence proportion (3.48% vs 0.639%) and prevalence (3.794% vs 0.708%) than the control group (n = 103,659,571). Following propensity score matching, 76,636 individuals per group were identified. Patients with OSA with PE demonstrated lower risks of cardiac arrest (RD −0.175%, RR 0.636 [CI, 0.539–0.751]; P < 0.0001) and all-cause mortality (RD −3.511%, RR 0.555 [CI, 0.533–0.579]; P < 0.0001) within one-month post-PE diagnosis compared to non-OSA patients with PE.

**Data availability statement:** We obtained the data for this study from the TriNetX platform, which provided the results as downloadable CSV files. We did not have any special access privileges; therefore, other researchers can replicate our cohorts using the inclusion criteria outlined in the Methods section. Legal restrictions apply to the availability of these data, as they were accessed under license for this study. Data related to this study are available from https://trinetx.com with permission from TriNetX.

**Funding:** The author(s) received no specific funding for this work.

**Competing interests:** Dr. Sharma declares receives grant and support for attending meetings or travel from West Virginia University (WVU) and National Science Foundation (NSF); SS has been on the speakers bureau of Zoll Respicardia Inc.(Until Nov 2024). SS has received Grant funding from INARI Medical Inc PEERLESS II trial, NIH RECOVER trial, rEST trial by Zoll Respicardia Inc. Dr. Alawad, Dr. Al-Saeed and Dr. Jarrar have no conflicts to declare. This does not alter our adherence to PLOS ONE policies on sharing data and materials.

**Abbreviations:** OSA, Obstructive sleep apnea; PE, Pulmonary embolism; VTE, Venous thromboembolism; DVT, Deep vein thrombosis.

## Conclusions

In the OSA population, the risk of PE is increased, while the probability of short-term mortality or cardiac arrest post diagnosis of PE is significantly low. These data suggest a complex relationship between OSA and the risk of PE and necessitate further evaluation of the potential mechanisms and clinical significance.

## Introduction

Obstructive sleep apnea (OSA) is a common and often underdiagnosed sleep disorder, affecting an estimated 25% of men and 10% of women [1]. Characterized by recurrent episodes of partial or complete obstruction of the upper airway during sleep, OSA leads to intermittent hypoxia, systemic inflammation, and other metabolic disturbances [2–5]. These pathophysiological mechanisms have been associated with an increased risk of thromboembolic events in OSA [6]. Recent research shows that patients with OSA could have a higher risk of an acute pulmonary embolism (PE), with a prevalence of 3.5% to 6.5%, compared to the general population, which has a prevalence of about 1% in patients [7,8].

A serious condition characterized by the obstruction of pulmonary arteries, and usually caused by thromboembolic events, PE can lead to significant outcomes, such as higher mortality, higher hospitalization rates, and need for critical care [9–12]. While the association between OSA and an elevated risk of PE has been suggested [7], the exact incidence of acute PE in individuals with OSA compared to the general population remains unclear. Furthermore, the outcome of acute PE in OSA patients has not been studied. Our primary hypothesis was that among patients with acute PE, those with OSA have higher 30-day mortality and cardiac arrest than adults without OSA. Secondary hypothesis included with greater intensive care needs (ICU) and endotracheal intubation in patients with acute PE and OSA compared to acute PE without OSA. By performing a multicenter retrospective study, we sought to understand the contributions of OSA to the management and eventual outcomes of PE and develop more precise clinical methods to evaluate a high risk patient population.

## Methodology

### Study design and data source

We carried out a retrospective cohort study through the TriNetX research network that collects de-identified electronic health record (EHR) data from numerous health care organizations in the United States and other countries [13]. TriNetX is a federated real-world data network that links healthcare organizations (HCOs)—including academic medical centers and hospitals—across multiple countries to create a harmonized repository of de-identified patient data for research and clinical investigation [13]. This research followed the Strengthening the Reporting of Observational Studies in Epidemiology (STROBE) guidelines. TriNetX de-identifies patient records based on the HIPAA Safe Harbor method by removing all 18 identifiers. Data are shared only in the aggregate, and because patients cannot be re-identified, institutional

review board oversight was not required for this analysis. Portions of the data that were available included demographics, comorbidities (ICD-9-CM and ICD-10-CM codes), procedures (CPT), medications (Veterans Affairs National Formulary), lab values (LOINC) and visit type. Data curation was done on August 16, 2025 (Fig 1).

## Cohort definition

The study population included adults (≥ 18 years) with an index acute PE event from January 1, 2013, to December 31, 2024, with 30 days of follow-up the outcomes after the PE incidence. There were two cohorts defined: (1) patients had a prior diagnosis of OSA and acute PE (OSA-PE cohort), and (2) patients had acute PE without prior OSA (PE-only cohort), these make the study population as defined under our inclusion criteria. OSA status was defined using ICD-10-CM G47.33, the ICD-9 codes are in Supplemental Table 1 in S1 File. PE was defined using ICD-10-CM I26 and the related ICD-9 codes, The data used from the years 2013−2015 employed the ICD-9-CM codes, which were then converted to ICD-10-CM equivalents, through the application of validated mapping algorithms [14,15]. Patients were excluded if they had other sleep-related breathing disorders such as central sleep apnea (ICD-10-CM G47.30; ICD-9 327.20) or alveolar hypoventilation (ICD-10-CM G47.30) to preserve diagnostic fidelity. We identified acute PE cases using the relevant ICD-9/ICD-10 diagnosis codes in both cohorts, without restricting to inpatient-specific encounter types, which may allow for identification of non-hospitalized cases even though acute PE almost always requires hospitalization.

## Outcomes

The two primary outcomes were all-cause mortality and cardiac arrest in the 30 days after the PE diagnosis. Patients were excluded from death and cardiac arrest if this occurred before the index PE event.

## Statistical analysis

The statistical analyses were performed using TriNetX Query Builder and Analytics. We used propensity score matching (1:1 nearest-neighbor) with a caliper of 0.1 of pooled standard deviations, and the maximum difference in propensity scores used for matching was 0.01. The match covariates were demographics (age, sex, race/ethnicity), comorbidities (hypertension, ischemic heart disease, heart failure, diabetes, cancer, liver disease, chronic lower lung diseases, including

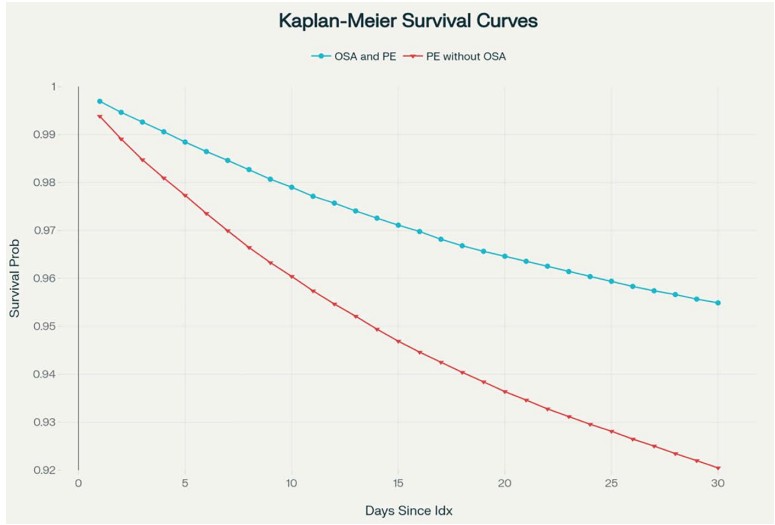

**Fig 1. Kaplan-Meier survival curves for all-cause mortality. Abbreviations:** PE = pulmonary embolism; OSA = obstructive sleep apnea.

COPD and asthma), body mass index category, smoking and alcohol use, medication classes [antiplatelet agents, anti-coagulants including direct oral anticoagulants (DOAC's), aspirin], and laboratory results (hemoglobin, platelets, INR, creatinine, bicarbonate). The balance was assessed using standardized mean differences, with < 0.1 acceptable.

Descriptive statistics were used to summarize baseline characteristics. Outcomes were expressed as incidence rates, relative risks (RRs), and risk differences (RDs) with 95% confidence intervals (CIs). Kaplan–Meier survival curves were produced for mortality and cardiac arrest outcomes, and log-rank tests were performed. The E-values were also calculated for effect estimates to examine sensitivity to unmeasured confounding.

## Results

### Study constituents

Within the entire cohort of 424,602 individuals with a diagnosis of PE, 92,461 had a prior OSA diagnosis and 332,141 did not. Before propensity score matching, patients with PE and OSA on average, were older (62.6 ± 13.9 vs 60.6 ± 17.9 years), they were more often male (53.5% vs 46.9%) and were more likely to be white (74.7% vs 69.5%) or non-Hispanic/Latino (80.6% vs 77.1%). They had substantially higher BMI (36.1 ± 9.6 vs 29.1 ± 7.3 kg/m²) and more than two-thirds were classified as obese (BMI ≥ 30) compared with approximately one-third of those without OSA. The cohort also had a more significant comorbidity burden, including more frequent hypertension, diabetes, chronic lower respiratory disease, ischemic heart disease, and venous disease. Again, consistent with these comorbidities, they had higher rates of medication use especially anticoagulant (65.2% vs 37.5%), but also antiplatelet therapy, including aspirin (40–42% vs~20%). Their laboratory values were consistent with subtle physiologic differences, higher bicarbonate levels, and somewhat higher hemoglobin and creatinine levels in the OSA group.

After performing 1:1 propensity score matching, each group had 76,636 individuals included in the analysis. Matching achieved excellent balance with respect to demographics, comorbidities, and medication use. The mean age was similar (62.7 vs 63.6 years) and the distribution of sex (~53% male), race, and ethnicity was almost identical. Cardiometabolic comorbidities and medication use were similar across groups. BMI remained mildly higher in the OSA group (34.5 vs 32.7 kg/m², SMD 0.21), with little difference for other clinical or laboratory metrics, providing additional evidence in support of the adequacy of covariate balance for later outcome comparisons (Table 1).

### Incidence proportion of PE

This analysis assessed the occurrence of PE from 1/1/2013 until 12/31/2024. The incidence proportion and prevalence of PE in individuals with OSA (n = 3,547,220) was compared to the general population without OSA (n = 103,659,571). An incidence proportion of PE in patients with OSA was 3.48% compared to 0.639% in the general population. and the prevalence was 3.794% in OSA group compared to 0.708% in general population (Fig 2).

### Primary outcomes

The two groups were followed for 30 days from the index event and assessed for the occurrence of cardiac arrest and all-cause mortality. In Group 1 (PE with OSA), 228 (0.3%) individuals had at least a single cardiac arrest event compared to 357 (0.5%) individuals in Group 2 (PE without OSA) (RD −0.175%, RR 0.636 [CI, 0.539–0.751]; P < 0.0001; E-value for RR 2.25) (Table 2 and Fig 3). The groups were then assessed for all-cause mortality with 3,340 (4.4%) in Group 1 compared to 5,979 (7.9%) in Group 2 (RD −3.511%, RR 0.555 [CI, 0.533–0.579]; P < 0.0001; E-value for RR 3.16) (Table 2 and Fig 3).

### Secondary outcomes

Secondary outcomes were defined as the requirement of critical care services, intubation occurrence, occurrence of gastrointestinal bleeding, and intracranial bleeding. Group 1 had fewer critical care admissions than Group 2, but the

**Table 1. Pre-matching and post-matching characteristics of Group 1(PE with OSA) and Group 2 (PE without OSA).**

| Characteristics | | Pre-matching | | | Post-matching | | | |
|---|---|---|---|---|---|---|---|---|
| | | PE in OSA (n=92 461) | PE without sleep OSA (n=332,141) | P Value | PE in OSA (n=76,636) | PE without OSA (n=76,636) | P Value | SMD |
| Age at Index | | 62.6±13.9 | 60.6±17.9 | < 0.0001 | 62.7±14.1 | 63.6±15.9 | < 0.0001 | 0.058 |
| Sex | Men | 49 430 (53.5%) | 155 905 (46.9%) | < 0.0001 | 40 630 (53.0%) | 40 955 (53.4%) | 0.096 | 0.008 |
| | Women | 43 013 (46.5%) | 175 914 (53.0%) | < 0.0001 | 35 992 (47.0%) | 35 586 (46.4%) | 0.038 | 0.011 |
| Race/Ethnicity | White | 69 056 (74.7%) | 230 713 (69.5%) | < 0.0001 | 56 579 (73.8%) | 57 006 (74.4%) | 0.013 | 0.013 |
| | Black or African American | 15 805 (17.1%) | 62 044 (18.7%) | < 0.0001 | 13 454 (17.6%) | 13 254 (17.3%) | 0.178 | 0.007 |
| | Not Hispanic or Latino | 74 509 (80.6%) | 256 191 (77.1%) | < 0.0001 | 61 516 (80.3%) | 61 557 (80.3%) | 0.792 | 0.001 |
| | Unknown Ethnicity | 13 547 (14.7%) | 55 826 (16.8%) | < 0.0001 | 11 377 (14.8%) | 11 392 (14.9%) | 0.914 | 0.001 |
| **Diagnosis** | | | | | | | | |
| Neoplasms | | 51,025 (41.7%) | 190,828 (31.8%) | <0.001 | 51,025 (41.7%) | 51,897 (42.4%) | <0.001 | 0.014 |
| Hypertensive diseases | | 65 447 (70.8%) | 117 903 (35.5%) | < 0.0001 | 50 324 (65.7%) | 50 698 (66.2%) | 0.044 | 0.01 |
| Ischemic heart diseases | | 30 882 (33.4%) | 44 539 (13.4%) | < 0.0001 | 22 198 (29.0%) | 22 241 (29.0%) | 0.809 | 0.001 |
| Other heart diseases | | 51 356 (55.5%) | 84 440 (25.4%) | < 0.0001 | 38 013 (49.6%) | 38 000 (49.6%) | 0.947 | < 0.0001 |
| Diabetes mellitus | | 34 381 (37.2%) | 46 475 (14.0%) | < 0.0001 | 23 908 (31.2%) | 23 766 (31.0%) | 0.433 | 0.004 |
| Chronic lower respiratory diseases | | 37 815 (40.9%) | 50 851 (15.3%) | < 0.0001 | 26 296 (34.3%) | 26 336 (34.4%) | 0.83 | 0.001 |
| Vein/lymphatic diseases | | 30 157 (32.6%) | 45 825 (13.8%) | < 0.0001 | 21 186 (27.6%) | 20 973 (27.4%) | 0.223 | 0.006 |
| Arterial diseases | | 22 336 (24.2%) | 34 135 (10.3%) | < 0.0001 | 16 009 (20.9%) | 16 040 (20.9%) | 0.846 | 0.001 |
| Neoplasms | | 38 194 (41.3%) | 92 825 (27.9%) | < 0.0001 | 30 126 (39.3%) | 30 029 (39.2%) | 0.612 | 0.003 |
| Blood & immune disorders | | 46 089 (49.8%) | 89 067 (26.8%) | < 0.0001 | 34 792 (45.4%) | 34 606 (45.2%) | 0.34 | 0.005 |
| Diseases of digestive system | | 62 840 (68.0%) | 124 624 (37.5%) | < 0.0001 | 48 445 (63.2%) | 48 512 (63.3%) | 0.723 | 0.002 |
| Liver diseases | | 14 928 (16.1%) | 23 682 (7.1%) | < 0.0001 | 10 655 (13.9%) | 10 491 (13.7%) | 0.224 | 0.006 |
| Nicotine dependence | | 15 229 (16.5%) | 34 151 (10.3%) | < 0.0001 | 11 708 (15.3%) | 11 586 (15.1%) | 0.385 | 0.004 |
| Alcohol-related disorders | | 4 937 (5.3%) | 11 253 (3.4%) | < 0.0001 | 3 821 (5.0%) | 3 771 (4.9%) | 0.556 | 0.003 |
| **Medications** | | | | | | | | |
| Anticoagulants | | 60 318 (65.2%) | 124 539 (37.5%) | < 0.0001 | 46 299 (60.4%) | 46 334 (60.5%) | 0.855 | 0.001 |
| Platelet aggregation inhibitors | | 38 996 (42.2%) | 68 186 (20.5%) | < 0.0001 | 28 893 (37.7%) | 29 008 (37.9%) | 0.545 | 0.003 |
| Aspirin | | 37 374 (40.4%) | 64 803 (19.5%) | < 0.0001 | 27 618 (36.0%) | 27 714 (36.2%) | 0.61 | 0.003 |
| **Laboratory** | | | | | | | | |
| BMI, mean±SD | | 36.1±9.6 | 29.1±7.3 | < 0.0001 | 34.5±9.3 | 32.7±8.1 | < 0.0001 | 0.214 |
| BMI category | 18.50 kg/m² | 2 394 (2.6%) | 10 922 (3.3%) | < 0.0001 | 2 032 (2.7%) | 1 915 (2.5%) | 0.059 | 0.01 |
| | 18.50-24.90 kg/m² | 9 404 (10.2%) | 56 265 (16.9%) | < 0.0001 | 9 087 (11.9%) | 8 712 (11.4%) | 0.003 | 0.015 |
| | 25.00-29.90 kg/m² | 20 986 (22.7%) | 75 819 (22.8%) | 0.404 | 18 998 (24.8%) | 19 050 (24.9%) | 0.758 | 0.002 |
| | 30.00-34.90 kg/m² | 27 343 (29.6%) | 55 473 (16.7%) | < 0.0001 | 21 844 (28.5%) | 21 808 (28.5%) | 0.839 | 0.001 |
| | 35.00-39.90 kg/m² | 24 692 (26.7%) | 29 134 (8.8%) | < 0.0001 | 16 093 (21.0%) | 15 923 (20.8%) | 0.285 | 0.005 |
| | ≥ 40.00 kg/m² | 25 232 (27.3%) | 18 766 (5.7%) | < 0.0001 | 13 852 (18.1%) | 13 482 (17.6%) | 0.014 | 0.013 |
| Hemoglobin, mean±SD | | 12.6±2.4 | 12.3±2.4 | < 0.0001 | 12.7±2.3 | 12.1±2.4 | < 0.0001 | 0.241 |
| Platelets, mean±SD | | 242.4±94.3 | 251.4±108.5 | < 0.0001 | 243.4±94.6 | 246.1±107.8 | < 0.0001 | 0.026 |
| INR, mean±SD | | 1.4±0.7 | 1.3±0.6 | < 0.0001 | 1.4±0.7 | 1.3±0.6 | < 0.0001 | 0.068 |
| Bicarbonate, mean±SD | | 26.0±3.9 | 25.5±3.6 | < 0.0001 | 25.9±3.8 | 25.5±3.7 | < 0.0001 | 0.107 |
| Creatinine, mean±SD | | 1.2±2.1 | 1.1±1.7 | < 0.0001 | 1.1±1.9 | 1.1±1.8 | 0.942 | <0.001 |

**Abbreviations:** OSA=obstructive sleep apnea, PE=pulmonary embolism, SMD=standardized mean difference, BMI=body mass index, INR=international normalized ratio.

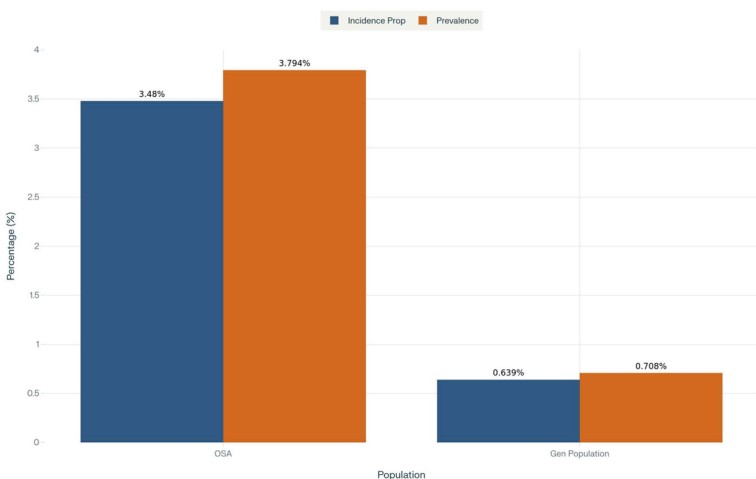

Incidence Proportion and Prevalence of PE

**Abbreviations:** PE = pulmonary embolism; OSA = obstructive sleep apnea

**Fig 2. Incidence proportion and prevalence of PE (2013-2024). Abbreviations:** PE = pulmonary embolism; OSA = obstructive sleep apnea.

**Table 2. Comparison of outcomes between PE in OSA patients versus PE in patients without OSA.**

| | Analysis | PE in OSA | PE in patients without sleep disorders | RD | RR (95% CI) | P value | E-value for RR | E-value of lower bound of 95% CI of RR |
|---|---|---|---|---|---|---|---|---|
| Primary outcomes | | | | | | | | |
| cardiac arrest | Before PSM | 337 (0.3%) | 1 189 (0.3%) | 0.017% | 1.052 (0.933-1.187) | 0.408 | 1.49 | 1.05 |
| | After PSM | 228 (0.3%) | 357 (0.5%) | −0.175% | 0.636 (0.539-0.751) | < 0.0001 | 2.25 | 1.84 |
| All-cause Mortality | Before PSM | 4 450 (4.3%) | 22 489 (5.9%) | −1.603% | 0.728 (0.706-0.752) | < 0.0001 | 2.34 | 2.25 |
| | After PSM | 3 340 (4.4%) | 5 979 (7.9%) | −3.511% | 0.555 (0.533- 0.579) | < 0.0001 | 3.16 | 3.03 |
| Secondary outcomes | | | | | | | | |
| Critical care services | Before PSM | 4 417 (5.5%) | 16 800 (5.1%) | 0.38% | 1.074 (1.040-1.110) | < 0.0001 | 1.27 | 1.14 |
| | After PSM | 3 349 (5.7%) | 3 488 (6.0%) | −0.217% | 0.964 (0.920-1.009) | 0.114 | 1.64 | 1.54 |
| Intubation | Before PSM | 1 754 (1.7%) | 5 915 (1.6%) | 0.176% | 1.112 (1.055-1.172) | < 0.0001 | 1.12 | 1 |
| | After PSM | 1 363 (1.8%) | 1 504 (2.0%) | −0.196% | 0.904 (0.841-0.972) | 0.006 | 1.8 | 1.63 |
| Gastrointestinal bleed | Before PSM | 958 (1.1%) | 3 992 (1.1%) | −0.052% | 0.953 (0.888-1.022) | 0.177 | 1.69 | 1.5 |
| | After PSM | 741 (1.1%) | 997 (1.5%) | −0.374% | 0.745 (0.678-0.819) | < 0.0001 | 2.5 | 2.25 |
| Intracerebral bleed | Before PSM | 167 (0.2%) | 902 (0.2%) | −0.075% | 0.686 (0.582-0.809) | < 0.0001 | 2.79 | 2.34 |
| | After PSM | 144 (0.2%) | 233 (0.3%) | −0.119% | 0.616 (0.501-0.759) | < 0.0001 | 3.08 | 2.45 |
| SDH | Before PSM | 126 (0.1%) | 509 (0.1%) | −0.011% | 0.803 (0.612-1.053) | 0.387 | 2.42 | 1.86 |
| | After PSM | 94 (0.1%) | 117 (0.2%) | −0.03% | 0.803 (0.612-1.053) | 0.112 | 2.85 | 1.86 |

**Abbreviations:** PSM = propensity score matching; SDH = subdural hematoma RR = relative risk; RD = risk difference; PE = pulmonary embolism; OSA = obstructive sleep apnea.

differences were not statistically significant [3,349 (5.7%) vs 3,488 (6.0%); RD −0.217%, RR 0.964 [CI, 0.920–1.009]; P = 0.114; E-value for RR 1.64]. Intubation was performed less often in Group 1 than in Group 2 [1,363 (1.8%) vs 1,504 (2.0%); RD −0.196%, RR 0.904 [CI, 0.841–0.972]; P = 0.006; E-value for RR 1.8]. Gastrointestinal bleeding occurred

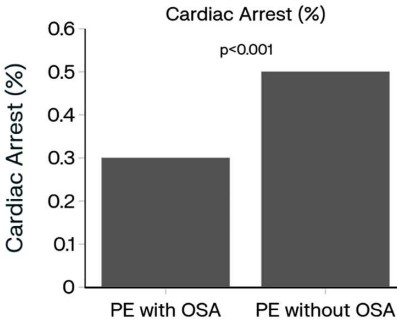 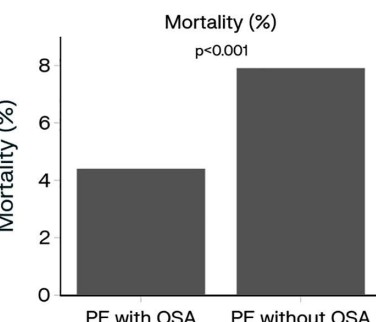

**Fig 3. Comparison 30-day cardiac arrest and mortality occurrence in OSA patients with acute PE versus without OSA. Abbreviations:** PE = pulmonary embolism; OSA = obstructive sleep apnea.

less often in Group 1 than in Group 2 [741 (1.1%) vs 997 (1.5%); RD −0.374%, RR 0.745 [CI, 0.678–0.819]; P < 0.0001; E-value for RR 2.5]. All forms of intracranial bleeding occurred less often in Group 1 than in Group 2 [144 (0.2%) vs 233 (0.3%); RD −0.119%, RR 0.616 [CI, 0.501–0.759]; P < 0.0001; E-value for RR 3.08].

## Discussion

This is the one of the largest propensity-matched studies to date assessing PE and its relation to OSA. A higher incidence (3.48%, 0.639%) of acute PE was observed in patients with OSA compared to those without Interestingly, despite this increased incidence and prevalence, patients with OSA experienced better outcomes per acute event of PE, with significantly lower rates of cardiac arrest and all-cause mortality within 30 days of the event. These findings suggest a potential influence of OSA on the clinical course of acute PE, although the underlying mechanisms remain uncertain.

This study identifies a significant association between OSA and an increased incidence of PE, with an incidence proportion of 3.794% in OSA patients compared to 0.64% in the general population without sleep disorders. These findings indicate that individuals with OSA are more than 6 times likely of having PE occur than those in the non-OSA general population. This finding is consistent with prior reports that OSA may be an important risk factor for thromboembolic events (DVT/PE) [6,7,16,17]. Moreover, Zhang et al observed that OSA patients had an elevated pulmonary artery obstruction index and PE severity scores, suggesting that OSA is associated with higher severity of PE [18]. The pathophysiological basis for thromboembolic risk is believed to be through hypoxia-induced platelet activation, coagulation cascade dysregulation, and endothelial dysfunction. OSA influences all components of Virchow's triad, leading to a hypercoagulable state and elevated VTE risk [17,19,20]. Alonso-Fernández et al (2020) also emphasized that intermittent hypoxia and sympathetic activation in OSA may accelerate the development of vascular thrombosis, making these patients more susceptible to PE [6].

Recent studies have revealed a complicated relationship between OSA and PE. Konnerth et al. documented acute PE as being more severe in those patients with OSA associated with higher mortality and prolonged hospital stay [8]. Conversely Ghaisi et al. reported that there was no increased risk of 30-day mortality in patients with PE and OSA as compared to those with PE without OSA [21]. These findings are inconsistent with our findings of lower 30-day mortality in patients with acute PE and OSA. However, Ghaisi et al.'s study was limited by a relatively small sample size, which could influence the results. In contrast, Huang et al. noted that OSA patients with PE had lower in-hospital mortality despite needing more ventilation [22]. Joshi et al. evaluated a large database and found that OSA was associated with lower in-patient mortality [23]. In keeping with this, a meta-analysis performed by Zhang et al. also found significantly lower in-hospital mortality [16]. Our study, which utilized a large database and is the only propensity-matched analysis specifically examining 30-day mortality and cardiac arrest, is supported by findings from prior smaller studies [7,23]. This

potential association with improved outcome may stem from the protective effect of chronic intermittent hypoxia, proposed by Alonso-Fernández et al. (2019) [6]. Nevertheless, OSA patients will still have an increased rate of long-term adverse events such as PE recurrence as discussed in studies that reported higher rates of PE recurrence underscoring the need for OSA screening in PE patients and the hypothetical benefits of treating OSA to modulate risk of PE recurrence [18,24].

We expand on the existing literature not just by evaluating outcome beyond in-hospital mortality to all-cause mortality by 30 days which gives a broader perspective to the prognosis of post-PE, but we also provide stratification of risk for cardiac arrest which was significantly greater for patients with acute PE without comorbid OSA. It is important to note that there was no difference shown in the requirements for intensive care unit service between those with OSA and those without OSA (5.7% versus 6.0%; P = 0.114). This suggests that even though the OSA patients had lower rates of 30-day mortality and cardiac arrest, their immediate critical care requirements were similar to the patients without OSA.

This study also revealed an unexpected finding: higher rates of gastrointestinal bleeding, intracerebral hemorrhage, and sub-dural bleeds in PE occurred in individuals without OSA compared to those with OSA and PE. This observation is similar to reports of obese stroke patients having less bleeding complications after thrombolytics compared to nonobese stroke patients (most likely due to lower weight-adjusted thrombolytic doses) [25]. Our findings align with some studies that have noted the paradoxical effects of OSA in patients with PE. For instance, Jiang et al observed that patients with OSA required higher doses of warfarin and had increased PE recurrence after anticoagulation discontinuation [26]. This supports the hypothesis that chronic intermittent hypoxia in OSA might lead to adaptive changes in the coagulation system, potentially reducing bleeding risks in PE [27].

These findings highlight the complex interplay between OSA and acute PE. While OSA appears to positively influence against 30-day mortality, cardiac arrest, and intubation rate, it has also shown increased risk of PE occurrence and recurrence. This underscores the importance of early OSA diagnosis and treatment in patients who have experienced acute PE [7]. further studies are needed to determine if early intervention with positive airway therapy would reduce recurrence.

## Strengths and limitation

The strength of the study is the examination of both a large dataset in a real-world setting and of several key outcomes based on a 30-day post-event period and not limited to hospitalization alone. Elsewhere, this study has several limitations that should be considered when interpreting the results. First, the dataset, specifically, had no documented follow-up duration limits in relation of outcomes against defined time points. That said, median follow-up was 28 days for the OSA cohort and 27 days for the general cohort. Second, despite propensity score matching, small baseline differences persisted between groups, which, given the large sample size, may have been statistically significant and potentially influenced associated measures. Third, the study is susceptible to unmeasured confounders, such as OSA severity, hypoxemic burden, polysomnography versus clinical OSA diagnosis, and pulmonary embolism severity (massive, submassive, segmental etc). Additionally, information is not available on positive airway pressure therapy in these patients and how it may have influenced outcome, although historically the adherence to PAP therapy has been low. Furthermore, In addition, while our findings indicate a potentially "protective" relationship, we caution against reading too much into this observation due to the nature of our study and missing key data about important factors such as CPAP treatment. Obesity is an independent risk factor for pulmonary embolism and is also associated with OSA [28]. However, the extent to which OSA contributes additional risk beyond that conferred by obesity remains unclear. Fourth, a residual BMI difference persisted between the groups (34.5 vs. 32.7 kg/m²; SMD = 0.214) despite propensity score matching. Although both values fall within the same obesity class, this imbalance could introduce confounding, given the independent association between obesity and PE risk as well as prognosis. Fifth, The study period (2013–2024) encompasses multiple advancements and shifts in PE management, including direct oral anticoagulants, catheter-directed therapies, and new criteria and protocols for risk stratification. We are unable to account for temporal bias. Finaly, limitations of the TriNetX platform prevented calculating cause-specific mortality and cardiac arrest and precluded the use of cluster-robust standard errors to account for patient clustering. Finally, as the study relied on ICD and procedure codes, its conclusions depend on consistent and accurate physician

coding. Coding bias may be introduced if coding practices vary across healthcare organizations, including differences in how conditions are classified or managed in inpatient versus outpatient settings.

## Conclusion

This study contributes to our understanding of the complex relationship between OSA and PE. While OSA increases the risk of PE occurrence, it appears to be potentially associated with better outcome against severe outcomes like 30-dat mortality, cardiac arrest and bleeding complications. These findings demonstrate the need for a multidimensional approach to treating PE in patients with OSA, and emphasize the importance of further investigating methods of care for this patient group.

## Supporting information

**S1 File. Propensity Score Matching for OSA and without OSA Groups with Acute PE.** Supplement table: Codes. (DOCX)

**S2 File. PE in OSA Query key for TriNetX.**
(DOCX)

## Author contributions

**Conceptualization:** Saud Alawad, Nawaf Al-Saeed, Ahmad Jarrar, Sijin Wen, Sunil Sharma.

**Data curation:** Saud Alawad, Nawaf Al-Saeed, Ahmad Jarrar, Sijin Wen, Sunil Sharma.

**Formal analysis:** Saud Alawad, Nawaf Al-Saeed, Ahmad Jarrar, Sijin Wen, Sunil Sharma.

**Funding acquisition:** Saud Alawad, Nawaf Al-Saeed, Ahmad Jarrar, Sijin Wen, Sunil Sharma.

**Investigation:** Saud Alawad, Nawaf Al-Saeed, Ahmad Jarrar, Sijin Wen, Sunil Sharma.

**Methodology:** Saud Alawad, Nawaf Al-Saeed, Ahmad Jarrar, Sijin Wen, Sunil Sharma.

**Project administration:** Saud Alawad, Nawaf Al-Saeed, Ahmad Jarrar, Sijin Wen, Sunil Sharma.

**Resources:** Saud Alawad, Nawaf Al-Saeed, Ahmad Jarrar, Sijin Wen, Sunil Sharma.

**Software:** Saud Alawad, Nawaf Al-Saeed, Ahmad Jarrar, Sijin Wen, Sunil Sharma.

**Supervision:** Saud Alawad, Nawaf Al-Saeed, Ahmad Jarrar, Sijin Wen, Sunil Sharma.

**Validation:** Saud Alawad, Nawaf Al-Saeed, Ahmad Jarrar, Sijin Wen, Sunil Sharma.

**Visualization:** Saud Alawad, Nawaf Al-Saeed, Ahmad Jarrar, Sijin Wen, Sunil Sharma.

**Writing – original draft:** Saud Alawad, Nawaf Al-Saeed, Ahmad Jarrar, Sijin Wen, Sunil Sharma.

**Writing – review & editing:** Saud Alawad, Nawaf Al-Saeed, Ahmad Jarrar, Sijin Wen, Sunil Sharma.

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
