## [Decision Letter · Decision Letter 0]

22 Oct 2025

Dear Dr. Sharma,

Thank you for submitting your manuscript to PLOS ONE. After careful consideration, we feel that it has merit but does not fully meet PLOS ONE’s publication criteria as it currently stands. Therefore, we invite you to submit a revised version of the manuscript that addresses the points raised during the review process.

We look forward to receiving your revised manuscript.

Kind regards,

Kartikeya Rajdev, MD

Academic Editor

PLOS ONE

Journal Requirements:

“Dr. Sharma declares receives grant and support for attending meetings or travel from West Virginia University (WVU) and National Science Foundation (NSF); SS has been on the speakers bureau of Zoll Respicardia Inc.(Until Nov 2024). SS has received Grant funding from INARI Medical Inc PEERLESS II trial, NIH RECOVER trial, rEST trial by Zoll Respicardia Inc.; and holds patents with WVU (invention numbers: US2023/0122156) that has been licensed by Premash Inc. SS is consultant for Premash Inc.

Dr. Alawad, Dr. Al-Saeed and Dr. Jarrar have no conflicts to declare”

3. Thank you for uploading your study's underlying data set. Unfortunately, the repository you have noted in your Data Availability statement does not qualify as an acceptable data repository according to PLOS's standards.

4. We notice that your supplementary [figures/tables] are included in the manuscript file. Please remove them and upload them with the file type 'Supporting Information'. Please ensure that each Supporting Information file has a legend listed in the manuscript after the references list.

Reviewers' comments:

Reviewer's Responses to Questions

**Comments to the Author**

1. Is the manuscript technically sound, and do the data support the conclusions?

Reviewer #1: Yes

Reviewer #2: Yes

Reviewer #3: Yes

Reviewer #4: Partly

2. Has the statistical analysis been performed appropriately and rigorously?

Reviewer #1: Yes

Reviewer #2: Yes

Reviewer #3: Yes

Reviewer #4: Yes

3. Have the authors made all data underlying the findings in their manuscript fully available?

Reviewer #1: Yes

Reviewer #2: Yes

Reviewer #3: Yes

Reviewer #4: Yes

4. Is the manuscript presented in an intelligible fashion and written in standard English?

Reviewer #1: Yes

Reviewer #2: Yes

Reviewer #3: Yes

Reviewer #4: Yes

Reviewer #1: No OSA severity (AHI index) or PE severity (massive/submassive/subsegmental) available. This limitation should be explicitly emphasized in the Major Limitations since severity likely affects mortality

Absence of CPAP/PAP adherence information weakens mechanistic interpretation this should be major not a minor limitation.

OSA appears to positively influence against 30-day mortality- consider revising to association language (“OSA was associated with lower observed 30-day mortality)

Reviewer #2: The study conclusion is some what paradoxical it does add some valuable information in terms of complex interplay between OSA and PE. The authors rightly acknowledge the strengths and limitations especially OSA severity, PE severity and anticoagulation regimens. Also since the study period is 2013-2024 where there is significant increase in usage of DOACs there is no information about this in the study.

Reviewer #3: I enjoyed reviewing this manuscript. Despite study limitations, as indicated in the manuscript, this study provides some valuable piece of information about the complex relationship between OSA and PE. It would have really been helpful to know the relationship between PAP compliance and PE.

Reviewer #4: This manuscript presents a large, contemporary TriNetX-based analysis evaluating how obstructive sleep apnea (OSA) influences the risk and short-term outcomes of acute pulmonary embolism (PE). The topic is timely and clinically relevant. The study adds novelty by including both incidence data and 30-day post-PE outcomes, particularly cardiac arrest, which extends prior inpatient-only reports such as Joshi et al., Pulmonary Circulation 2021. The dataset spans 2013–2024, reflecting modern therapeutic practices including DOACs and catheter-directed interventions. However, several methodological and reporting issues limit interpretability. There are internal inconsistencies in sample sizes between the Abstract and Results, residual BMI imbalance after propensity matching, unclear encounter definitions (hospitalized vs outpatient PE), and incomplete details on mortality ascertainment. The question is important and the data are strong, but the paper requires a major revision before it can be considered further.

**Do you want your identity to be public for this peer review?** For information about this choice, including consent withdrawal, please see our Privacy Policy

Reviewer #1: No

Reviewer #2: **Yes:** naga praneeth raja

Reviewer #3: **Yes:** Shais Shafi Jallu

Reviewer #4: **Yes:** Sivaramakrishna Cheetirala

---

## [Author Response · Author response to Decision Letter 1]

4 Nov 2025

PONE-D-25-51061

Obstructive sleep apnea and outcomes in acute pulmonary embolism: a large-scale database study

PLOS ONE

Dear Dr. Sharma,

Thank you for submitting your manuscript to PLOS ONE. After careful consideration, we feel that it has merit but does not fully meet PLOS ONE’s publication criteria as it currently stands. Therefore, we invite you to submit a revised version of the manuscript that addresses the points raised during the review process.

We look forward to receiving your revised manuscript.

Kind regards,

Kartikeya Rajdev, MD

Academic Editor

PLOS ONE

Journal Requirements:

“Dr. Sharma declares receives grant and support for attending meetings or travel from West Virginia University (WVU) and National Science Foundation (NSF); SS has been on the speakers bureau of Zoll Respicardia Inc.(Until Nov 2024). SS has received Grant funding from INARI Medical Inc PEERLESS II trial, NIH RECOVER trial, rEST trial by Zoll Respicardia Inc.; and holds patents with WVU (invention numbers: US2023/0122156) that has been licensed by Premash Inc. SS is consultant for Premash Inc.

Dr. Alawad, Dr. Al-Saeed and Dr. Jarrar have no conflicts to declare”

Response: Confirmed and included

Response: Done

3. Thank you for uploading your study's underlying data set. Unfortunately, the repository you have noted in your Data Availability statement does not qualify as an acceptable data repository according to PLOS's standards.

Response: The data used in this study were accessed through the TriNetX Research Network, a global federated health research platform. The data comprise aggregated, de-identified patient information that cannot be shared or downloaded in raw form per the data use agreement and network governance policies.

Researchers interested in replicating or validating these findings can request access to TriNetX through institutional membership or by contacting TriNetX (https://www.trinetx.com) to obtain access to the same analytic tools and datasets under a similar data use agreement. As per TriNetX policy and data use agreements, the underlying patient-level data cannot be shared or deposited in a public repository. However, all analyses were conducted within the TriNetX platform using reproducible query criteria, which are attached as a supplement.

4. We notice that your supplementary [figures/tables] are included in the manuscript file. Please remove them and upload them with the file type 'Supporting Information'. Please ensure that each Supporting Information file has a legend listed in the manuscript after the references list.

Done

Noted. None was added

Reviewers' comments:

Reviewer's Responses to Questions

Comments to the Author

1. Is the manuscript technically sound, and do the data support the conclusions?

Reviewer #1: Yes

Reviewer #2: Yes

Reviewer #3: Yes

Reviewer #4: Partly

2. Has the statistical analysis been performed appropriately and rigorously?

Reviewer #1: Yes

Reviewer #2: Yes

Reviewer #3: Yes

Reviewer #4: Yes

3. Have the authors made all data underlying the findings in their manuscript fully available?

Reviewer #1: Yes

Reviewer #2: Yes

Reviewer #3: Yes

Reviewer #4: Yes

4. Is the manuscript presented in an intelligible fashion and written in standard English?

Reviewer #1: Yes

Reviewer #2: Yes

Reviewer #3: Yes

Reviewer #4: Yes

5. Review Comments to the Author

Reviewer #1: No OSA severity (AHI index) or PE severity (massive/submassive/subsegmental) available. This limitation should be explicitly emphasized in the Major Limitations since severity likely affects mortality

Response: We thank the reviewer for the comments. We have already mentioned OSA severity and PE severity in the limitations section. As per the suggestions we have added (massive/ submassive/ subsegmental) to the sentence)

Absence of CPAP/PAP adherence information weakens mechanistic interpretation this should be major not a minor limitation.

Response: We believe we have adequately stressed the limitation of lack of CPAP adherence data- “ Furthermore, In addition, while our findings indicate a potentially "protective" relationship, we caution against reading too much into this observation due to the nature of our study and missing key data about important factors such as CPAP treatment”

OSA appears to positively influence against 30-day mortality- consider revising to association language (“OSA was associated with lower observed 30-day mortality)

Response :We appreciate the reviewer comments and agree with the above suggestion. We have revised the language as recommended.

Reviewer #2: The study conclusion is some what paradoxical it does add some valuable information in terms of complex interplay between OSA and PE. The authors rightly acknowledge the strengths and limitations especially OSA severity, PE severity and anticoagulation regimens. Also since the study period is 2013-2024 where there is significant increase in usage of DOACs there is no information about this in the study.

Response: We than the review for their astute observation. We however did include DOAC’s in the analysis. This is mentioned now more explicitly in the methods section.

Reviewer #3: I enjoyed reviewing this manuscript. Despite study limitations, as indicated in the manuscript, this study provides some valuable piece of information about the complex relationship between OSA and PE. It would have really been helpful to know the relationship between PAP compliance and PE.

Response: We are glad that the reviewer enjoyed the manuscript. We agree that the study provides important insight into the relationship between PAP and PE. Unfortunately the PAP compliance information is not included in the TriNetX database, otherwise we would have gladly included that data.

Reviewer #4: This manuscript presents a large, contemporary TriNetX-based analysis evaluating how obstructive sleep apnea (OSA) influences the risk and short-term outcomes of acute pulmonary embolism (PE). The topic is timely and clinically relevant. The study adds novelty by including both incidence data and 30-day post-PE outcomes, particularly cardiac arrest, which extends prior inpatient-only reports such as Joshi et al., Pulmonary Circulation 2021. The dataset spans 2013–2024, reflecting modern therapeutic practices including DOACs and catheter-directed interventions. However, several methodological and reporting issues limit interpretability. There are internal inconsistencies in sample sizes between the Abstract and Results, residual BMI imbalance after propensity matching, unclear encounter definitions (hospitalized vs outpatient PE), and incomplete details on mortality ascertainment. The question is important and the data are strong, but the paper requires a major revision before it can be considered further.

Response: We thank the reviewers for their comments. We have rechecked and there are no internal inconsistencies between abstract and the results. We do acknowledge that the data required more explanation i.e. clearly stating that outcome data was based on propensity matching and not overall sample size. This has been clarified in the abstract.

Response: The residual BMI imbalance has been addressed in the limitations section. As noted, although a difference in BMI persisted despite propensity matching, both groups remained within the same obesity class. This has been explicitly acknowledged in the limitations.

By default, acute pulmonary embolism is typically managed in the inpatient setting. However, it is possible that a subset of patients in the sample may have been treated in ambulatory settings. Because the ICD code is the same for both, the dataset does not allow differentiation between inpatient and outpatient management. This clarification has also been incorporated into the limitations section. Mortality data was limited to all-cause mortality. Mortality language has also been changed as suggested from “ positively influencing” to “OSA was associated with lower observed 30-day mortality”

6. PLOS authors have the option to publish the peer review history of their article (what does this mean?). If published, this will include your full peer review and any attached files.

If you choose “no”, your identity will remain anonymous but your review may still be made publi

---

## [Decision Letter · Decision Letter 1]

14 Dec 2025

Dear Dr. Sharma,

Thank you for submitting your manuscript to PLOS ONE. After careful consideration, we feel that it has merit but does not fully meet PLOS ONE’s publication criteria as it currently stands. Therefore, we invite you to submit a revised version of the manuscript that addresses the points raised during the review process.

We look forward to receiving your revised manuscript.

Kind regards,

Kartikeya Rajdev, MD

Academic Editor

PLOS One

**Journal Requirements:**

Reviewers' comments:

Reviewer's Responses to Questions

**Comments to the Author**

Reviewer #1: All comments have been addressed

Reviewer #2: (No Response)

Reviewer #3: All comments have been addressed

Reviewer #4: All comments have been addressed

2. Is the manuscript technically sound, and do the data support the conclusions?

Reviewer #1: Yes

Reviewer #2: Yes

Reviewer #3: Yes

Reviewer #4: Yes

3. Has the statistical analysis been performed appropriately and rigorously?

Reviewer #1: Yes

Reviewer #2: Yes

Reviewer #3: Yes

Reviewer #4: Yes

4. Have the authors made all data underlying the findings in their manuscript fully available?

Reviewer #1: Yes

Reviewer #2: Yes

Reviewer #3: Yes

Reviewer #4: Yes

5. Is the manuscript presented in an intelligible fashion and written in standard English?

Reviewer #1: Yes

Reviewer #2: Yes

Reviewer #3: Yes

Reviewer #4: Yes

Reviewer #1: It is well written paper however the limitation are pertinent to the type of data you are extracting. I have noticed the suggestion revisions are appropriately addressed

Reviewer #2: lack of inpatient vs outpatient differentiation of PE as ICD codes frequently capture chronic PE, follow up evaluation in the out patient setting.

Also with out PE severity stratification it is some what premature to assume that there is physiological protection from obesity.

Reviewer #3: (No Response)

Reviewer #4: Thank you for the revised submission and for the thoughtful responses to the reviewer comments. The authors have adequately addressed the previously raised concerns, including clarification of cohort definitions, refinement of associative language, expansion of the limitations section to appropriately acknowledge unavailable data on OSA severity, pulmonary embolism severity, and PAP adherence, and clarification regarding anticoagulation strategies, mortality ascertainment, and ICD-based encounter definitions. The revisions have improved the clarity, transparency, and interpretability of the manuscript without altering its core findings

**Do you want your identity to be public for this peer review?** For information about this choice, including consent withdrawal, please see our Privacy Policy

Reviewer #1: **Yes:** Anil Kumar Nalla

Reviewer #2: No

Reviewer #3: **Yes:** Shais Jallu MD FCCP

Reviewer #4: **Yes:** sivaramakrishna cheetirala

---

## [Author Response · Author response to Decision Letter 2]

2 Jan 2026

Reviewer #1: It is well written paper however the limitation are pertinent to the type of data you are extracting. I have noticed the suggestion revisions are appropriately addressed

We appreciate the reviewers comments

Reviewer #2: lack of inpatient vs outpatient differentiation of PE as ICD codes frequently capture chronic PE, follow up evaluation in the out patient setting.

Also with out PE severity stratification it is some what premature to assume that there is physiological protection from obesity.

Finally, as the study relied on ICD and procedure codes, its conclusions depend on consistent and accurate physician coding. Coding bias may be introduced if coding practices vary across healthcare organizations, including differences in how conditions are classified or managed in inpatient versus outpatient settings.

Furthermore, In addition, while our findings indicate a potentially "protective" relationship, we caution against reading too much into this observation due to the nature of our study and missing key data about important factors such as CPAP treatment.

Reviewer #3: (No Response)

Reviewer #4: Thank you for the revised submission and for the thoughtful responses to the reviewer comments. The authors have adequately addressed the previously raised concerns, including clarification of cohort definitions, refinement of associative language, expansion of the limitations section to appropriately acknowledge unavailable data on OSA severity, pulmonary embolism severity, and PAP adherence, and clarification regarding anticoagulation strategies, mortality ascertainment, and ICD-based encounter definitions. The revisions have improved the clarity, transparency, and interpretability of the manuscript without altering its core findings

We appreciate the reviewers comments

---

## [Decision Letter · Decision Letter 2]

29 Jan 2026

Obstructive sleep apnea and outcomes in acute pulmonary embolism: a large-scale database study

PONE-D-25-51061R2

Dear Dr. Sharma,

We’re pleased to inform you that your manuscript has been judged scientifically suitable for publication and will be formally accepted for publication once it meets all outstanding technical requirements.

Kind regards,

Kartikeya Rajdev, MD

Academic Editor

PLOS One

Additional Editor Comments (optional):

Reviewers' comments:

Reviewer's Responses to Questions

**Comments to the Author**

Reviewer #2: All comments have been addressed

2. Is the manuscript technically sound, and do the data support the conclusions?

Reviewer #2: Yes

3. Has the statistical analysis been performed appropriately and rigorously?

Reviewer #2: Yes

4. Have the authors made all data underlying the findings in their manuscript fully available?

Reviewer #2: Yes

5. Is the manuscript presented in an intelligible fashion and written in standard English?

Reviewer #2: (No Response)

Reviewer #2: The authors carefully reviewed the comments and carefully addressed them. The article is now thoroughly written.

**Do you want your identity to be public for this peer review?** For information about this choice, including consent withdrawal, please see our Privacy Policy

Reviewer #2: **Yes:** Naga Praneeth Raja

---

## [Editor Report · Acceptance letter]

PONE-D-25-51061R2

PLOS One

Dear Dr. Sharma,

I'm pleased to inform you that your manuscript has been deemed suitable for publication in PLOS One. Congratulations! Your manuscript is now being handed over to our production team.

Kind regards,

on behalf of

Dr. Kartikeya Rajdev

Academic Editor

PLOS One